# Long-Term Alcohol Consumption Caused a Significant Decrease in Serum High-Density Lipoprotein (HDL)-Cholesterol and Apolipoprotein A-I with the Atherogenic Changes of HDL in Middle-Aged Korean Women

**DOI:** 10.3390/ijms23158623

**Published:** 2022-08-03

**Authors:** Kyung-Hyun Cho, Hyo-Seon Nam, Dae-Jin Kang, Min-Hee Park, Ju-Hyun Kim

**Affiliations:** 1Raydel Research Institute, Medical Innovation Complex, Daegu 41061, Korea; sun91120@rainbownature.com (H.-S.N.); daejin@rainbownature.com (D.-J.K.); mhpark@rainbownature.com (M.-H.P.); aksk1694@rainbownature.com (J.-H.K.); 2LipoLab, Yeungnam University, Gyeongsan 712749, Korea

**Keywords:** alcohol, apolipoproteins A-I, high-density lipoproteins, low-density lipoproteins, paraoxonase

## Abstract

Light-to-moderate alcohol drinking is associated with a low incidence of cardiovascular disease (CVD) via an elevation of high-density lipoproteins-cholesterol (HDL-C), particularly with the short-term supplementation of alcohol. However, there is no information on the change in the HDL qualities and functionalities between non-drinkers and mild drinkers in the long-term consumption of alcohol. This study analyzed the lipid and lipoprotein profiles of middle-aged Korean female non-drinkers, mild-drinkers, and binge-drinkers, who consumed alcohol for at least 10 years. Unexpectedly, the serum levels of HDL-C and apolipoprotein A-I (apoA-I) were decreased significantly depending on the alcohol amount; the binge-drinker group showed 18% and 13% lower HDL-C (*p* = 0.011) and apoA-I levels (*p* = 0.024), respectively, than the non-drinker group. Triglyceride (TG) and oxidized species, malondialdehyde (MDA), and low-density lipoproteins (LDL) levels were significantly elevated in the drinker groups. Interestingly, the binge-drinker group showed 1.4-fold higher (*p* = 0.020) cholesterol contents in HDL_2_ and 1.7-fold higher (*p* < 0.001) TG contents in HDL_3_ than those of the non-drinker group. The mild-drinker group also showed higher TG contents in HDL_3_ (*p* = 0.032) than the non-drinker group, while cholesterol contents were similar in the HDL_3_ of all groups. Transmission electron microscopy (TEM) showed that the non-drinker group showed a more distinct and clear particle shape of the LDL and HDL image with a larger particle size than the drinker group. Electrophoresis of LDL showed that the drinker group had faster electromobility with a higher smear band intensity and aggregation in the loading position than the non-drinker group. The HDL level of binge drinkers showed the lowest paraoxonase activity, the highest glycated extent, and the most smear band intensity of HDL and apoA-I, indicating that HDL quality and functionality were impaired by alcohol consumption. In conclusion, long-term alcohol consumption in middle-aged women, even in small amounts, caused a significant decrease in the serum HDL-C and apoA-I with atherogenic changes in LDL and HDL, such as an increase in TG and MDA content with a loss of paraoxonase activity.

## 1. Introduction

Alcohol consumption is commonly associated with a disturbance of lipid metabolism, fat accumulation in the liver, hepatic steatosis, and hepatic cirrhosis. Alcoholic fatty liver is often associated with hyperlipidemia and alcoholic hepatitis [1]. The mechanisms of the lipid accumulation are increased cholesterol synthesis in the liver and decreased release of serum lipoproteins [2]. Although ethanol and its metabolic derivative, such as acetaldehyde, have been classified as carcinogenic to humans by the International Agency for Research on Cancer [3], moderate ethanol consumption has been reported to suppress the incidence of cardiovascular disease (CVD) [4], but the mechanism is still not completely understood.

Meta-analysis has shown that moderate alcohol drinking is inversely associated with the incidence of CVD via the elevation of HDL-C [5]. Consumption of 40.9 g/day of alcohol for 4.1 weeks resulted in a 5.1 mg/dL increase in HDL-C from 25 studies [5]. A study with Korean males also showed an increase in HDL-C in the drinker group, around 10% higher than the non-drinker group. Drinking was associated with an increased risk of metabolic syndrome [6]. On the other hand, another report linked 30 g of alcohol consumption with a 24.7% reduction in the risk of CVD from the projected reduction in three biomarkers: HDL-C, fibrinogen, and TG [5]. The anti-atherogenic effect of moderate alcohol consumption was linked with an increase in HDL-C and a decrease in insulin resistance [7]. Although light-to-moderate alcohol drinking is associated with a lower incidence of CVD via an increase in HDL-C [8], there are no reports proving that the effects can be applied to all genders and ages.

The protective effect of alcohol might not be applicable to all genders and ages because men and women have different lipid metabolisms and alcohol degradation capacities [9,10]. On the other hand, although drinkers usually showed higher serum HDL-C levels than non-drinkers, there is a positive association between alcohol consumption and the risk of metabolic syndrome [11,12]. Metabolic syndrome is a cluster of conditions including abdominal obesity, elevated blood pressure, hyperglycemia, elevated TG, and low serum HDL-C.

There has been no sufficient information on the long-term changes in HDL-C in female drinkers because many reports on the effects of short-term and additive drinking on the HDL-C level have focused mainly on male participants or a promiscuous group [13]. The current study was designed to compare the qualities of HDL and LDL between drinkers and non-drinkers, particularly in middle-aged women, because there are no reports on the change in HDL and LDL quality depending on alcohol consumption. The increased effect of HDL-C by alcohol can differ according to gender, age, and alcohol consumption period because the quantity of HDL-C can change during a lifetime [14].

The HDL-C level is not fixed during one’s lifetime. In particular, middle-aged women show a dramatic decrease in HDL-C around 50 years old due to menopause [14]. Hence, it has not been firmly established that the protective effect of alcohol in suppressing CVD can be applied to all ages and genders because men and women have different lipid metabolisms and changes in HDL-C levels during their lifetime. Furthermore, there has been limited number of studies comparing the change in HDL quality between absolute non-drinkers and drinkers with long-term alcohol consumption [15,16]. Comparison of the HDL quality is also essential for evaluating the anti-atherogenic effects of alcohol consumption because the HDL quality determines the LDL quality to protect oxidation and glycation from the production of oxidized LDL (oxLDL) and small-dense LDL (sdLDL).

The alcohol elimination rate varies according to gender, age, race, and dietary habits. Furthermore, lipid and lipoprotein metabolism differ depending on gender and age [17,18]. On the other hand, there is little information on the change in HDL and LDL quantity and quality by long-term alcohol consumption depending on gender. The current study was designed to compare the quality of HDL and LDL among non-drinkers, mild drinkers, and binge drinkers, especially in middle-aged women who consumed alcohol for at least 10 years of adulthood.

## 2. Results

### 2.1. Anthropometric Profiles and Blood Parameters

As shown in Table 1, all participants were divided into three groups depending on the extent of alcohol consumption: absolute non-drinkers (group 1), mild-drinkers (group 2, 50 ± 6 g of alcohol per month), and binge-drinkers (group 3, 493 ± 106 g of alcohol per month). The three groups showed a similar age, around 50.1 ± 1.0 years old distribution of 35–64 years (Table 1). No remarkable differences in the anthropometric data, such as BMI, blood pressure, muscle mass, fat mass, and handgrip strength (HS), were observed among the groups. All participants showed a normal range of anthropometric profiles of middle-aged women and were healthy without diagnosed diseases (Table 1).

In the blood lipid profiles, as shown in Table 2, the binge-drinker group showed an 18% lower HDL-C level than the non-drinker group (*p* = 0.011), but there was no difference in the TC and TG levels among the groups. The binge-drinker group showed the smallest %HDL-C/TC level, highest TG level, and TG/HDL-C level, but significance was not detected in the between-group comparison. Interestingly, there was no difference in the LDL-C among the groups around 150 mg/dL, but the binge-drinking group showed the highest LDL-C/HDL-C. The binge-drinker group showed a 13% lower apoA-I level than the non-drinker group, as shown in Table 2. Although there was no significance, the binge-drinker group showed the highest apo-B level and apo-B/apoA-I ratio. The mild-drinker group and non-drinker group showed similar HDL-C and apoA-I. These results suggest that quantities of HDL-C and apoA-I are decreased by binge drinking, not mild drinking.

Among the three groups, although there was no difference in serum total cholesterol (TC) level, the LDL-C, glucose, hsCRP, and hepatic enzyme levels, such as AST, ALT, and γ-GTP, were higher in the drinker groups. Although no significance was detected between any group comparisons, the binge-drinking group showed an approximately two-fold higher γ-GTP level than the non-drinker and mild-drinker groups. These results suggest that binge drinking (daily 16 ± 3 g of alcohol) in middle-aged women did not cause notable inflammatory parameters in the blood.

### 2.2. Characteristics of Lipoproteins Depend on Alcohol Intake

In LDL, the drinker group (groups 2 and 3) showed significantly elevated oxidized species, malondialdehyde (MDA), levels around 1.4-fold higher than the non-drinking group, as shown in Table 3. The drinker group, groups 2 and 3, showed a 21% and 26%, respectively, smaller LDL size (nm^2^) than the non-drinker group (group 1) with significance and dependent on the alcohol intake (Table 3). Although there was no difference in the TC content in LDL among the three groups, the TG content in LDL was remarkably elevated in the drinker group (*p* = 0.004). Post hoc analysis showed that groups 2 and 3 had 1.4-fold (*p* = 0.027) and 1.5-fold (*p* = 0.005) higher TG levels, respectively, than group 1. On the other hand, there was no difference in the particle size and glycation extent in LDL among the three groups. These results suggest that drinking caused the accumulation of oxidized species and TG content in LDL, even though a small amount of alcohol was consumed.

The mild-drinker and binge-drinker groups showed 33% (*p* = 0.007) and 35% (*p* = 0.048) lower HDL_2_ associated with the PON activity than the non-drinker group; the reduction extent of the PON activity was similar regardless of the extent of drinking. In HDL_2_, only the TC content was elevated significantly in group 3, which was 1.3-fold higher than in group 1 (*p* = 0.020), even though the TG content was similar in the three groups. In addition, there was no difference in the particle size and glycation extent in the HDL_2_ among the groups. These results suggest that features of HDL_2_ were impaired by alcohol drinking regardless of alcohol amount; a small amount of alcohol can impair the HDL quality and functionality, especially by long-term consumption.

In HDL_3_, however, the particle size and PON activity were decreased significantly by alcohol consumption even though the effect was not dose-dependent. Groups 2 and 3 showed 32% (*p* < 0.001) and 25% smaller (*p* = 0.008) particle sizes, respectively, than group 1. On the other hand, the TG content was 1.3-fold (*p* = 0.032) and 1.5-fold (*p* < 0.001) higher in groups 2 and 3, respectively, than in group 1. Although there was no difference in the TC content, the TG content was elevated with a concomitant decrease in the HDL size and PON activity, as shown in Table 3.

### 2.3. LDL Particle and Lipid Composition

Structural and compositional analysis of LDL revealed no significant differences in the LDL particle size, glycated extent, and cholesterol content in LDL among the three groups. On the other hand, both drinker groups showed significantly higher TG contents and oxidized species (expressed in malondialdehyde, MDA) in LDL than the non-drinker group (Table 3 and Figure 1). Interestingly, the TG content in LDL from the drinking groups increased depending on the level of alcohol consumption, as shown in Table 3; the mild-drinker and binge-drinker groups showed 1.4-fold (*p* = 0.027) and 1.5-fold (*p* = 0.005) higher TG contents in LDL, respectively, than the non-drinker group.

As shown in Figure 1A, LDL electrophoresis on agarose gel showed that the non-drinker group had the strongest and distinct band intensity without aggregation in the loading position (lane 1). In contrast, the mild drinker group showed the fastest electromobility with two split bands and smear band intensity (lane 2). The binge-drinker group (lane 3) showed a larger smear band intensity with aggregation at the loading position. Oxidized LDL by cupric ion treatment (lane 4) showed the fastest electromobility with the largest smear band intensity and aggregation at the loading position, as indicated by the red arrowhead. Interestingly, mild drinkers and binge drinkers showed 52% and 33%, respectively, smaller band intensity and more aggregated band at the loading position. More oxidized LDL was prone to be aggregated in the loading position due to increased apo-B fragmentation in LDL with the fastest electromobility, as indicated by the blue arrow and red arrowhead in Figure 1A. There was no difference in LDL electromobility between the mild drinker and binge drinker, suggesting that oxidative damage in LDL was similar regardless of the drinking amount in long-term consumption.

Quantification of oxidized species (MDA level) in LDL showed that MDA in LDL was higher in the drinker group but not in a dose-dependent manner. The mild-drinker group showed 1.5-fold (*p* = 0.013) higher level of oxidized species in LDL (1.7 nM of MDA) than the non-drinker group (1.1 nM of MDA), as shown in Table 3 and Figure 1B. Binge-drinker group (group 3) showed a 1.4-fold higher content of MDA in LDL (1.6 nM of MDA) than the non-drinker group (group 1), suggesting that long-term alcohol intake, even though a small amount of alcohol, could induce the same extent of oxidative damage in LDL.

As shown in Figure 2, TEM showed that the non-drinker group (group 1, photo a) had a distinct particle shape and clear morphology with a 25–26 nm diameter, while the mild-drinker (group 2, photo b) and binge-drinker (group 3, photo c) groups showed more ambiguous morphology with 23–25 nm diameter. The binge-drinker group (482 ± 28 nm^2^) and mild-drinker group (507 ± 22 nm^2^) showed a smaller LDL size than the non-drinker group (643 ± 36 nm^2^). OxLDL by the cupric ion treatment showed the smallest particle size and a loss of particle morphology with severe aggregation (photo d).

### 2.4. Comparison of HDL Particles and Lipid Composition

In HDL_2_, the non-drinkers (group 1, photo a) showed the most distinct particle shape and clear morphology, but the drinker group showed an ambiguous particle shape (Group 2, photo b) and aggregated pattern (Group 3, photo c), as shown in Figure 3. There was a remarkable difference in the particle size of HDL_2_ among the groups; the mild-drinker group (group 2) showed the smallest particle size, 121 ± 6 nm^2^, while the non-drinker (group 1) and binge-drinker (group 3) groups showed particle sizes of 214 ± 15 nm^2^ and 186 ± 13 nm^2^, respectively. Interestingly, the binge-drinker group (group 3) showed the highest TC content in HDL_2_, 1.3-fold higher than group 1 (*p* = 0.020), even though the TG content was similar. The glycation extent in HDL_2_ increased with increasing alcohol intake; the binge-drinker group showed the greatest glycation extent among the three groups, 8% higher than the non-drinker group (Table 3). Agarose electrophoresis without denaturation showed that HDL_2_ from the binge-drinker group (lane 3) had the fastest electromobility with the greatest smear band intensity similar to that of glycated HDL (lane g), while the non-drinkers showed the slowest electromobility with the most distinct band intensity. Interestingly, the TC content in HDL_2_ was higher in the drinker groups (*p* = 0.052) in a dose-dependent manner. The binge-drinker group exhibited the highest TC content in HDL_2_, 1.4-fold higher than that of the non-drinker group (*p* = 0.020).

In HDL_3_, the binge-drinker group showed the highest TG content (*p* = 0.002), but TC content was similar in the three groups. The mild-drinker and binge-drinker groups showed a 1.3-fold (*p* = 0.032) and 1.7-fold (*p* < 0.001) higher TG content in HDL_3_, respectively, than the control group. The binge-drinker group showed the lowest PON activity (*p* = 0.010). The non-drinker group showed a 1.6-fold higher PON activity than the binge-drinker group. The mild-drinker and non-drinker groups showed similar PON activity in HDL_3_, even though the mild-drinker group had a higher TG content than the non-drinker group. These results suggest that the TG content in HDL_3_ increased according to the alcohol intake amount, while the PON activity did not.

As shown in Figure 4, SDS-PAGE showed that HDL_3_ from the binge-drinker group had a 15% smaller apoA-I band intensity, as indicated by the blue arrow with a slightly shifted up band than the non-drinker and mild-drinker groups. This is a typical band pattern of glycated apoA-I with more smear and multimerization band pattern as indicated by red arrowheads. HDL_3_-associated PON activity was impaired severely in the binge-drinking group, approximately 37% smaller than the non-drinking group (*p* = 0.012). These results suggest that alcohol consumption impaired the HDL quality and functionality via multimerization of apoA-I, and the loss of paraoxonase activity depends on the alcohol intake amount.

TEM image analysis showed that the non-drinker group had the largest HDL_3_ particle size of 113 ± 41 nm^2^ with a distinct particle shape, while glycated HDL_3_ showed the smallest particle size, 57 ± 22 nm^2^, with a more aggregated and fibrous morphology, as shown in Figure 5. The mild-drinker and binge-drinker groups showed a smaller range of particle sizes, 78 ± 15 nm^2^ and 85 ± 28 nm^2^, respectively, with more unclear morphology than group 1. These results suggest that drinking caused a decrease in HDL particle size and more aggregation despite the small amounts of long-term alcohol consumption.

### 2.5. Spearman Correlation Analysis

Spearman correlation analysis with all participants revealed a significant negative correlation between HDL-C (r = −0.365, *p* = 0.007) and apoA-I (r = −0.274, *p* = 0.045) and the alcohol intake amount, as shown in Table 4. Although the serum TG and apo-B, and γ-GTP showed no difference from the group comparisons (Table 2), there were significant positive correlations of TG (r = 0.262, *p* = 0.005), TG/HDL-C (r = 0.493, *p* < 0.001), apo-B/apoA-I ratio (r = 0.364, *p* = 0.007), and γ-GTP (r = 0.581, *p* < 0.001) in all participants, as shown in Table 4. These results suggest that the serum TG and apo-B, pro-atherogenic parameters, could be increased by alcohol consumption in a dose-dependent manner. Although γ-GTP was elevated by alcohol consumption in a dose-dependent manner, other hepatic inflammatory parameters, AST, ALT, and hs-CRP, were not significantly correlated. On the other hand, AST (r = 0.132) and ALT (r = 0.114) were slightly positively associated, and hs-CRP showed a negative association (r = −0.037) without significance.

In LDL, however, there was no linear correlation between the LDL-MDA level (r = 0.071, *p* = 0.608) and alcohol intake (Table 4), even though the mild-drinker group showed significantly higher LDL-MDA levels than the non-drinker group (Table 3). The LDL-TC and LDL-TG were negatively (r = −0.079) and positively (r = 0.132) correlated, respectively, but the differences were not significant (Table 4)

In HDL, the PON activity in HDL_3_ (PON-HDL_3_) was negatively correlated with the increase in alcohol consumption (r = −0.439, *p* = 0.022), while the TG content in HDL_3_ (HDL_3_-TG) was positively correlated (r = 0.211, *p* = 0.141). Interestingly, TC and TG in HDL and TG in LDL were positively correlated with the alcohol consumption level except for LDL-TC (r = −0.079), even though they showed no significance. These results suggest that the quality of HDL was impaired in the drinker group regardless of drinking extent, especially in the TG content.

## 3. Discussion

Alcohol consumption is associated with a moderate elevation of HDL-C, but the elevation effect resulted mainly from short-term supplementation, around 2–6 weeks [19], and additional consumption of alcohol by volunteers from both genders. Alcohol consumption through two weeks of supplementation of 1 mL vodka/kg/day produced an 18% increase in HDL-C by raising the transport rates of apoA-I [20]. On the other hand, the effects of alcohol on lipoprotein metabolism have been investigated mainly in men. Forty-two studies showed that an alcohol intake of less than 100 g per day increased the HDL-C to a greater extent in men than women [5]. Regarding the elevation of HDL-C, men showed a 1.4-fold higher coefficient for a 1 g increase in alcohol (b = 0.134 mg/dL) than women (b = 0.095 mg/dL). ApoA-I was also increased up to 11.8 mg/dL after 3.9 weeks of 37.6 g of alcohol per day. A meta-analysis with 24 studies revealed a 0.294 mg/dL increase in apoA-I per gram per day of alcohol [5]. On the other hand, the elevation effect of HDL-C and apoA-I were observed mainly in studies with men and short-term supplementation of alcohol [19,21,22]. Furthermore, there is no information on the change in HDL quality and functionality, which were increased by the short-term consumption of alcohol. Moreover, there has been no study to explain the long-term effects of alcohol consumption for at least ten years in middle-aged women regarding the change in HDL-C, HDL functionality, and qualities of HDL and LDL.

The major findings of this study are the decrease in HDL-C and apoA-I with increasing alcohol intake (Table 2). At the lipoprotein level, however, there was no dose dependency of alcohol intake on the increase in oxidation extent and TG content in LDL and the decrease in HDL_3_ particle size between groups 2 and 3 (Table 3). The serum HDL-C and apoA-I levels showed significant negative associations with alcohol intake, while the serum apoA-I ratio, TG, and γ-GTP showed significant positive associations (Table 4). Among the lipoprotein profiles, the HDL_3_-associated PON activity showed a negative association significantly with alcohol intake (Table 4), indicating that antioxidant activity was impaired by alcohol in a dose-dependent manner. Concomitantly, the drinker group (Group 3) showed significantly greater oxidation extent in LDL (Figure 1) and smaller HDL_2_ and HDL_3_ particle sizes than those in the non-drinker group (Figure 2 and Figure 4). The drinker groups showed a smaller particle size of HDL_2_ with a more ambiguous morphology and smear band intensity than the non-drinker group (Figure 3). The HDL_3_ particles from the drinker group also showed more multimerization and loss of PON activity (Figure 4) with smaller HDL_3_ particle size (Figure 5) than the non-drinker group. These results suggest that long-term drinking caused the modification of HDL and LDL to have atherogenic properties dependent on the alcohol intake in middle-aged women.

Unexpectedly, the current results were opposite those of many reports, which showed that moderate drinking (intake of 30 g per day) could elevate the HDL-C by about 3.8 mg/dL from a meta-analysis with 25 experimental studies [19,21]. Nonetheless, those studies focused only on the change in HDL-C quantity; they did not compare the HDL quality and functionality between drinkers and non-drinkers. Because HDL has potent antioxidant and anti-inflammatory activities in blood with cholesterol efflux activity to suppress the initiation of atherogenesis, maintaining good quality HDL is vital for HDL functionality. A higher apoA-I content, no multimerization and HDL aggregation, and a larger particle size are typical features of good HDL quality. On the other hand, the drinker group of middle-aged women showed a decrease in apoA-I (Table 2), an increase in the extent of LDL oxidation and smaller LDL size (Table 3 and Figure 1), modification of apoA-I (Figure 2), impairment of antioxidant ability (Figure 3), and a smaller HDL particle size (Figure 4). These impairments of HDL functionality were related to the detrimental change in LDL, elevation of MDA, and smaller particle size, to have the typical features of oxidized LDL

The current results contradict a previous report showing that atheroprotective change in the lipid and lipoprotein profiles occurred after the short-term consumption of alcohol in relatively younger women [23]. Thirty-seven premenopausal women aged from 21 to 40 years consumed mild amounts of alcohol (two drinks containing approximately 30 g per day for three months), which resulted in a 10% increase in HDL (both HDL_2_ and HDL_3_) and an 8% decrease in LDL. The HDL particle size increased with a concomitant 7% increase in the apoA-I content in HDL. Although there were ethnic and age differences between studies, these discrepancies might have originated from the long-term and short-term consumption of alcohol. In contrast to previous studies, which were designed to supplement additional alcohol, the current study conducted a retrospective evaluation of the changes in LDL and HDL quality after the long-term consumption of alcohol for at least 10 years. Similarly, the long-term consumption of alcohol (21 years) in middle-aged men and women aged 30–59 years resulted in a remarkable decrease in cholesterol efflux and uptake abilities of HDL [24], indicating severe impairment of HDL functionality.

Alcohol intake was associated with an increase in triglyceridemia [25,26]. The elevation of serum TG is associated with an increase in the TG content in HDL, which is linked to the production of dysfunctional HDL. Indeed, the serum TG level was elevated in group 3, and the TG content was increased in HDL_3_, not in HDL_2_. Interestingly, the triglyceride-rich HDL_3_ from patients with familial hypercholesterolemia showed more pro-inflammatory properties with a decreased cholesterol efflux ability [27]. In addition, elevated TG content in HDL diminished the capacity of CE delivery via scavenger receptor class B type I (SR-BI) [28]. LDL and HDL enriched in triglyceride promoted abnormal cholesterol transport [29] and an increased risk of myocardial infarction and ischemic stroke [30]. Overall, the elevated TG contents in HDL and LDL contribute to the progression of the atherogenic and inflammatory processes.

On the other hand, moderate alcohol intake is associated with decreased oxidative stress, reduction of blood pressure, and increased HDL-C, while excessive alcohol intake is associated with hypertension, type 2 diabetes, and increased serum TG [13]. Nevertheless, the effects of alcohol are not the same for all people; they might differ depending on gender, amount of alcohol consumed, duration, and patterns of intake (occasional, daily, or binge) [31]. Moreover, women exhibited greater sensitivity to the toxic effects of alcohol than men with decreased metabolism over the same amount of alcohol [32]. Long-term heavy alcohol drinking is related to the development of cardiovascular disorders (arrhythmias, cardiomyopathy, and stroke) or cancer (breast, esophagus, mouth, colon, and rectum) [33].

Although light-to-moderate alcohol drinking has been linked with the beneficial effects of reducing the mortality risk of CVD, CHD, and stroke, it cannot be applied to all gender and ages. The current results suggested that the qualities of HDL, such as the antioxidant activity of paraoxonase, and LDL, such as the oxidation extent, can be impaired by long-term alcohol consumption in middle-aged Korean women who consume even small amounts of alcohol. Although the binge-drinker group (47.6 ± 1.6 years old) was relatively younger than the other groups, they showed the lowest serum HDL-C and apoA-I levels and HDL-PON activity with the highest TG content in HDL. More interestingly, the mild drinkers showed the highest LDL-MDA level and higher LDL-TG level than the non-drinker group, even though they showed similar serum HDL-C and apoA-I levels and HDL_3_-PON activity to the drinker group.

A limitation of this study was the data on alcohol consumption obtained from questionnaires. The validation of these data for alcohol intake extent was very intricate. Another concern is the unequal distribution of menopausal women among the groups; groups 2 and 3 had the largest number (*n* = 12) and smallest number (*n* = 4) of menopausal women, respectively. This unequal distribution of menopausal status between groups might interfere interpretation of the current results. Because menopausal women displayed more atherogenic lipid and lipoprotein profiles with increased dysfunctional HDL, it might explain why group 2 showed more oxidized LDL and smaller HDL_2_ particle size than group 3. On the other hand, an interesting finding of this study was that the serum HDL-C and apoA-I levels in middle-aged women were decreased by a small amount of drinking, while the TC and LDL-C were relatively unchanged.

In conclusion, middle-aged women with long-term alcohol consumption, even in small amounts, showed a decrease in the serum levels of HDL-C, apoA-I, and HDL-associated paraoxonase activity with an increase in LDL oxidation and TG content in LDL and HDL. These results suggest that atherogenic changes in the lipid and lipoprotein profiles might be associated with alcohol consumption in middle-aged women (35–62 years old).

## 4. Materials and Methods

### 4.1. Participants

Female middle-aged (35–62 years old) volunteers were recruited randomly by a nationwide advertisement in Korea from 2021 to 2022. The participants were divided into three groups depending on their alcohol consumption per month: non-drinkers, mild drinkers, and binge drinkers. This study was approved by the Korea National Institute for Bioethics Policy (KoNIBP, approval number P01-202109-31-009) by the Ministry of Health Care and Welfare (MOHW) of Korea. The alcohol intake was estimated from a self-administered questionnaire inquiring about the frequency and amount of alcohol consumed per month. Non-drinkers had abstained from any alcoholic beverages. Mild drinkers (group 2, 21–105 g of alcohol per month) were those who have consumed a drink containing alcohol in the last 12 months, according to the WHO definition (https://www.who.int/data/gho/indicator-metadata-registry/imr-details/458 (accessed on 20 July 2022)), who consume less than one unit (15 g of alcohol) of drinking per week, around 50 ± 6 g of alcohol per month. Binge drinking was defined as consuming five or more drinks on one occasion for men or four or more drinks on one occasion for women. Binge-drinkers (group 3, 168–1680 g of alcohol per month) were defined as four units (60 g of alcohol) of drinking per week, around 493 ± 106 g of alcohol per month. Based on the questionnaire, the participants in groups 2 and 3 began drinking at 19 years old, which is around graduation from high school. Since the youngest participants in groups 2 and 3 were 35 years old, it is reasonable to expect they consumed alcohol at least 10 years after graduation from high school.

### 4.2. Anthropometric Analysis

The blood pressure was measured using an Omron HBP-9020 (Kyoto, Japan). The height, body weight, body mass index (BMI), body water, total body fat (%), total body fat mass (kg), and visceral fat mass (VFM) (kg) were measured individually using an X-scan plus II body composition analyzer (Jawon Medical, Gyeongsan, Korea). The handgrip strength (HS) was measured in the standing position with the arms straight down to the sides. The maximum grip strengths of the right and left hands were measured three times alternatively using a digital hand dynamometer (digital grip strength dynamometer, T.K.K 5401; Takei Scientific Instruments Co., Ltd., Tokyo, Japan). After the handgrip strengths of both hands were measured, a 60-second resting interval was allowed. The maximum grip strength of the dominant hand was used for the analysis [34].

### 4.3. Blood Analysis

After overnight fasting, blood was collected using a vacutainer (BD Bio Sciences, Franklin Lakes, NJ, USA) without the addition of an anti-coagulant. The serum parameters in Table 2 were determined by an automatic analyzer using Cobas C502 chemistry analyzer (Roche, Germany) at a commercially available diagnostic service via SCL healthcare (Seoul, Korea).

### 4.4. Isolation of Lipoproteins

Very low-density lipoproteins (VLDL, d < 1.019 g/mL), LDL (1.019 < d < 1.063), HDL_2_ (1.063 < d < 1.125), and HDL_3_ (1.125 < d < 1.225) were isolated from an individual patient and control sera via sequential ultracentrifugation [35], with the density adjusted by the addition of NaCl and NaBr in accordance with standard protocols. The samples were centrifuged for 24 h at 10 °C at 100,000× *g* using a Himac NX (Hitachi, Tokyo, Japan) at the Raydel Research Institute (Daegu, Korea).

For each of the lipoproteins purified individually, the total cholesterol (TC) and TG measurements were obtained using commercially available kits (cholesterol, T-CHO, and TG, Cleantech TS-S; Wako Pure Chemical, Osaka, Japan). The protein concentrations of the lipoproteins were determined using the Lowry protein assay, as modified by Markwell et al. [36] using the Bradford assay reagent (Bio-Rad, Seoul, Korea) with bovine serum albumin (BSA) as a standard.

### 4.5. LDL Oxidation and Quantification

To assess the degree of oxidation of individual LDL was assessed by measuring the concentration of oxidized species in LDL according to the thiobarbituric acid reactive substances (TBARS) method using malondialdehyde (MDA) as a standard [36].

Oxidized LDL (oxLDL) was produced by incubating the LDL fraction with CuSO_4_ (final concentration, 10 μM) for 4 h at 37 °C. The oxLDL was then filtered (0.2 μm) and analyzed using a thiobarbituric acid reacting substances (TBARS) assay to determine the extent of oxidation, as described elsewhere [37].

### 4.6. Paraoxonase Assay

The paraoxonase-1 (PON-1) activity toward paraoxon was determined by evaluating the hydrolysis of paraoxon into *p*-nitrophenol and diethylphosphate, which was catalyzed by the enzyme [37]. PON-1 activity was then determined by measuring the initial velocity of *p*-nitrophenol production at 37 °C, as determined by measuring the absorbance at 415 nm (microplate reader, Bio-Rad model 680; Bio-Rad, Hercules, CA, USA), as described previously [38,39].

### 4.7. Electromobility of Lipoproteins

The electromobility of the participants’ samples was compared by evaluating the migration of each lipoprotein (LDL, HDL_2_, and HDL_3_) evaluated by agarose electrophoresis [40]. The relative electrophoretic mobility depends on the intact charge and three-dimensional structure of HDL. Hence, agarose gel electrophoresis was carried out with HDL_2_ and HDL_3_ from each group under the non-denatured state [41]. The gels were then dried and stained with 0.125% Coomassie Brilliant Blue, after which the relative band intensities were compared via band scanning using Gel Doc ^®^ XR (Bio-Rad) with Quantity One software (version 4.5.2) of Bio-Rad (Hercules, CA, USA).

### 4.8. Glycation of HDL

The purified HDL (2 mg/mL) was incubated with 250 mM D-fructose in 200 mM potassium phosphate/0.02% sodium azide buffer (pH 7.4) for up to 72 hr in air containing 5% CO_2_ at 37 °C. Fructose can induce a remarkably greater glycation extent of apoA-I than glucose, according to a previous report [42]. The extent of glycation was determined by reading the fluorometric intensity at 370 nm (excitation) and 440 nm (emission), as described previously [43], using an LS55 spectrofluorometer (PerkinElmer) and a 1 cm path-length suprasil quartz cuvette (Fisher Scientific, Pittsburg, PA, USA).

### 4.9. Electron Microscopy

Transmission electron microscopy (TEM, Hitachi H-7800; Ibaraki, Japan) located at Raydel Research Institute (Daegu, Korea) was performed at an acceleration voltage of 80 kV. HDL was negatively stained with 1% sodium phosphotungstate (PTA; pH 7.4) with a final apolipoprotein concentration of 0.3 mg/mL in TBS. Then, 5 μL of the HDL suspension was blotted with filter paper and replaced immediately with a 5 μL droplet of 1% PTA. After a few seconds, the stained HDL fraction was blotted onto a Formvar carbon-coated 300 mesh copper grid and air-dried. The shape and size of the HDL were determined by TEM at a magnification of 40,000×, according to a previous report [41].

### 4.10. Data Analysis

All analyses in Table 1, Table 2, Table 3 and Table 4 were normalized by a homogeneity test of variances through Levene’s statistics. If not normalized, nonparametric statistics were used Kruskal–Wallis test. All values are expressed as the median (25%; 75%) for the continuous variables for the middle-aged women group. Multiple groups were compared using a one-way analysis of variance (ANOVA) in Table 1, Table 2 and Table 3. All tests were two-tailed, and the statistical significance was defined at *p* < 0.05.

In Table 1, we analyzed the anthropometric profiles of the middle-aged women group depending on alcohol intake. Anthropometric profiles, such as alcohol intake amount, height, weight, heart rate, SBP, DBP, muscle mass, fat mass, and body water, were compared using ANOVA, and age, fat mass_visceral, HS, and BMI were compared using Kruskal–Wallis test.

In Table 2, we analyzed the blood lipid and inflammatory parameters of the middle-aged women group depending on alcohol intake. The blood lipid and inflammatory parameters, such as HDL-C, HDL-C/TC, LDL-C, LDL-C/HDL-C, Apo-B, Glucose, hs-CRP, and AST, were compared using ANOVA, and TC, TG, TG/HDL-C, ApoA-I, Apo-B/ApoA-I, ALT, and g-GTP were compared using Kruskal–Wallis test.

In Table 3, we analyzed the characteristics of lipoproteins in the middle-aged women group depending on alcohol intake. The characteristics of lipoproteins, such as LDL-MDA, LDL-TG, HDL_2_-PON, HDL_3_-TC, and HDL_3_-PON, were compared using ANOVA, and LDL-size, LDL-glycation, LDL-TC, HDL_2_-size, HDL_2_-glycation, HDL_2_-TC, HDL_2_-TG, HDL_3_-size, and HDL_3_-TG were compared using Kruskal–Wallis test.

As a post hoc analysis, the Bonferroni test was used to determine the significance of the differences in the continuous variables to identify the differences between each group. Spearman rank correlation analysis was carried out to find a positive or negative association in Table 4. Statistical analyses were carried out using the SPSS statistical package version 28.0 (SPSS Inc., Chicago, IL, USA), incorporating sampling weights and adjusting for the complex survey design.

## Figures and Tables

**Figure 1 ijms-23-08623-f001:**
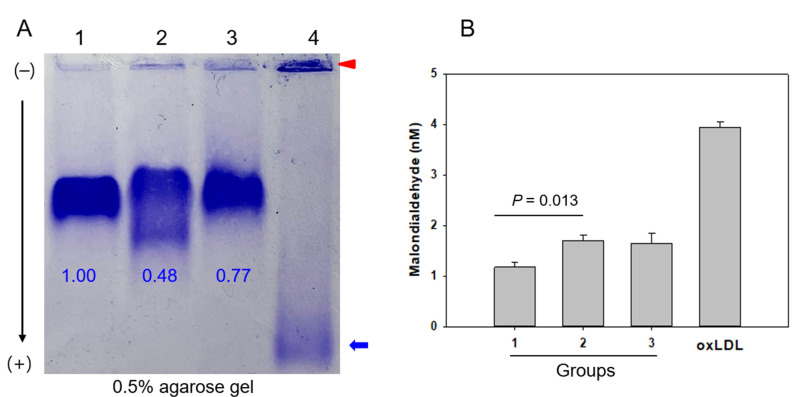
Native electrophoresis of LDL from each group and quantification of oxidized species. (**A**) Comparison of electromobility and aggregation extent of LDL (10 μg) band on 0.5% agarose gel without denaturation. Lane 1, group 1; lane 2, group 2; lane 3, group 3; lane 4, oxLDL, Cu^2+^ treated for 4 h. Red arrowhead indicates aggregated band at loading position. Blue arrow indicates faster electromobility of oxLDL with smear band intensity. (**B**) Quantification of oxidized species using malondialdehyde standard by the thiobarbituric acid reactive substance (TBARS) assay.

**Figure 2 ijms-23-08623-f002:**
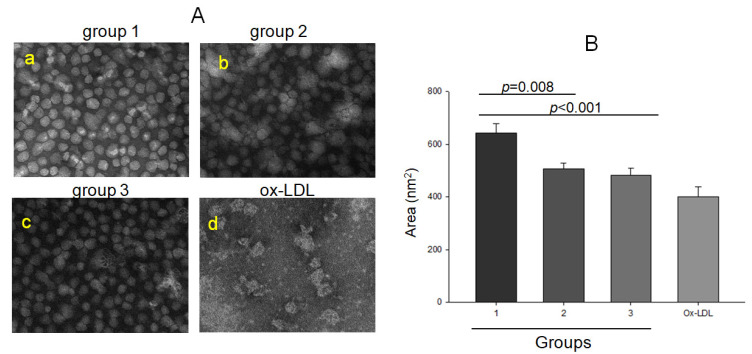
Transmission electron microscopy (TEM) images at a magnification with 40,000× (**A**) and area analysis (**B**) of LDL from each group.

**Figure 3 ijms-23-08623-f003:**
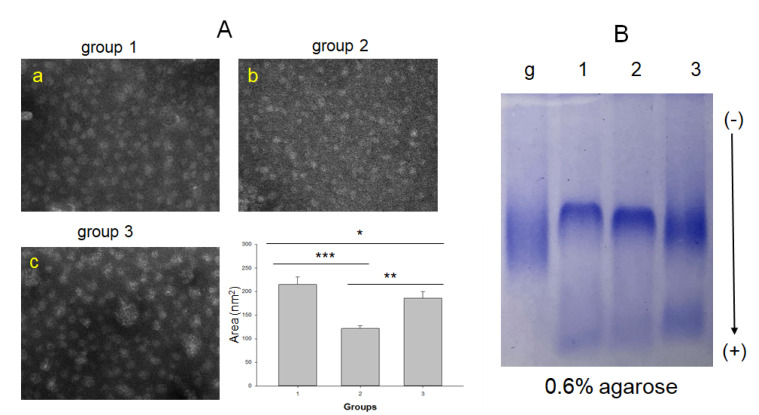
Transmission electron microscopy (TEM) images at a magnification with 40,000× and area analysis of HDL_2_ from each group (**A**) and comparison of the native state electromobility on 0.6% agarose gel (**B**). * *p* < 0.05; ** *p* < 0.01; *** *p* < 0.001. lane g, glycated HDL_2_; lane 1, group 1; lane 2, group 2; lane 3, group 3.

**Figure 4 ijms-23-08623-f004:**
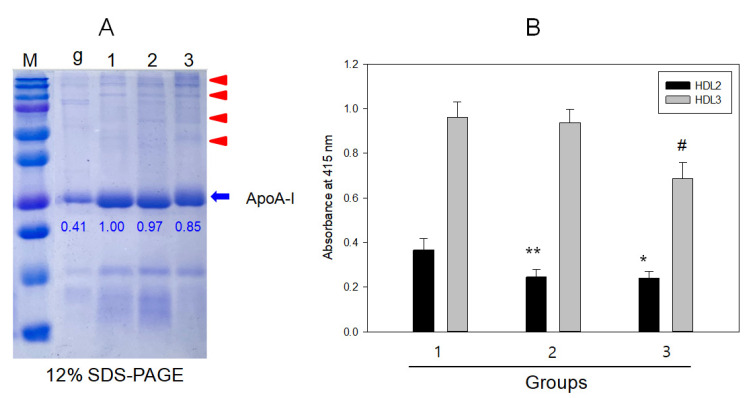
Electrophoretic patterns of HDL_3_ (**A**) and HDL-associated paraoxonase activity (**B**) from each group. Lane M, molecular weight marker, Precision plus protein^TM^ standards (Cat#161-0394, Bio-Rad); lane g, glycated HDL; lane 1, group 1; lane 2, group 2; lane 3, group 3. g-HDL was glycated HDL by fructose treatment (final 250 mM) for 72 h. The red arrow indicates a multimerized band.* *p* < 0.05 versus group 1; ** *p* < 0.01 versus group 1; # *p* < 0.05 versus group 1.

**Figure 5 ijms-23-08623-f005:**
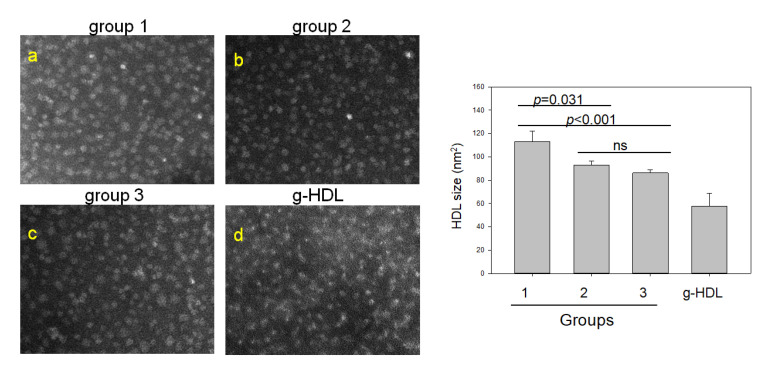
Transmission electron microscopy (TEM) image at a magnification with 40,000× of HDL_3_ from each group and the particle size distribution. The TEM image was obtained with a magnification of 40,000×. g-HDL, glycated HDL; ns—not significant.

**Table 1 ijms-23-08623-t001:** Anthropometric profiles of middle-aged women group depending on alcohol intake *.

	Group 1 (0 g) ^1^ *n* = 18	Group 2 (21~105 g) ^1^ *n* = 20	Group 3 (168~1680 g) ^1^ *n* = 16	*p*	1 vs. 2	1 vs. 3	2 vs. 3
Age (year-old), (min, max)	53.0 (45.0; 58.0) (38, 64)	52.0 (49.3; 55.8) (35, 62)	49.5 (41.3; 51.8) (36, 58)	0.163	1.000	0.208	0.427
Menopausal (n)	10	12	4				
Alcohol intake amount (g/month)	0	63 (21; 63) ^1^	315 (210; 560) ^2^	<0.001	1.000	<0.001	<0.001
Height (cm)	157.5 (156.0; 162.3)	160.5 (157.3; 162.8)	162.0 (157.3; 166.8)	0.147	1.000	0.174	0.468
Weight (kg)	60.3 (51.6; 62.6)	55.5 (50.7; 60.6)	58.6 (54.1; 61.1)	0.611	1.000	1.000	1.000
BMI (kg/m^2^), (min., max.)	23.7 (20.5; 24.9) (19.2, 26.4)	21.5 (20.2; 23.1) (18.7, 30.9)	21.5 (21.0; 23.7) (17.0, 29.3)	0.295	0.360	1.000	1.000
Heart rate (BPM)	76.0 (70.5; 86.3)	73.5 (70.8; 80.5)	72.0 (69.3; 82.5)	0.518	1.000	0.856	1.000
SBP (mmHg)	123.5 (116.8; 132.0)	119.0 (108.5; 123.3)	123.5 (114.5; 135.8)	0.044	0.090	1.000	0.103
DBP (mmHg)	68.5 (65.0; 77.0)	70.0 (65.5; 79.5)	77.0 (66.3; 84.0)	0.433	1.000	0.985	0.650
Muscle mass (kg)	37.7 (34.9; 40.2)	37.6 (34.7; 40.1)	38.4 (36.3; 40.5)	0.760	1.000	1.000	1.000
Fat mass_subcutaneous (kg)	16.6 (12.5; 17.2)	13.6 (11.2; 16.5)	14.3 (12.8; 16.2)	0.584	0.967	1.000	1.000
Fat mass_visceral (kg)	2.0 (1.3; 2.2)	1.5 (1.2; 1.9)	1.6 (1.4; 2.1)	0.341	0.430	1.000	1.000
handgrip strength (HS, kg) (min., max.)	27.5 (26.3; 29.3) (13, 33)	26.5 (23.0; 28.8) (19, 32)	26.0 (23.0; 29.0) (21, 31)	0.374	0.572	0.828	1.000
Body water (kg)	29.6 (27.8; 31.5)	29.4 (27.1; 31.3)	30.2 (28.4; 31.6)	0.650	1.000	1.000	1.000

BMI—body mass index; BPM—beat per minute; DBP—diastolic blood pressure; HS—handgrip strength; SBP—systolic blood pressure; SEM—standard error of the mean. ^1^ Range of alcohol consumption amount (g) per month 21~105 g in group 2; ^2^ range of alcohol consumption amount (g) per month 168~1680 g in group 3. * Data are presented as median (25th; 75th percentile).

**Table 2 ijms-23-08623-t002:** Blood lipid and inflammatory parameters of middle-aged women group depending on the alcohol intake *.

	Group 1 (0 g) *n* = 18	Group 2 (21~105 g) ^1^ *n* = 20	Group 3 (168~1680 g) ^2^ *n* = 16	*p*	1 vs. 2	1 vs. 3	2 vs. 3
TC (mg/dL)	219.5 (200.5; 239.2)	232.1 (185.6; 245.9)	216.4 (197.6; 246.8)	0.947	1.000	1.000	1.000
HDL-C (mg/dL)	56.3 (47.6; 61.6)	52.8 (47.7; 59.2)	45.2 (38.5; 55.5)	0.012	0.864	0.011	0.123
HDL-C/TC (ratio)	25.5 (21.8; 29.4)	24.1 (21.3; 28.6)	22.1 (17.6; 26.3)	0.096	1.000	0.106	0.366
LDL-C (mg/dL)	138.0 (126.5; 173.5)	158.5 (121.5; 174.5)	146.5 (132.5; 176.3)	0.864	1.000	1.000	1.000
TG (mg/dL)	84.5 (54.2; 104.6)	67.0 (56.4; 95.0)	108.2 (62.2; 167.4)	0.170	1.000	0.667	0.186
TG/HDL-C (ratio)	1.6 (0.8; 2.3)	1.2 (1.0; 1.7)	2.4 (1.3; 3.6)	0.088	1.000	0.273	0.105
LDL-C/HDL-C (ratio)	2.7 (2.1; 3.3)	2.9 (2.3; 3.5)	3.1 (2.7; 4.1)	0.123	1.000	0.127	0.646
ApoA-I (mg/dL)	181.5 (169.5; 197.5)	169.0 (156.0; 187.8)	165.5 (145.3; 182.3)	0.029	0.375	0.024	0.663
Apo-B (mg/dL)	101.5 (81.5; 126.3)	100.0 (81.5; 112.5)	107.5 (92.5; 130.5)	0.249	0.933	1.000	0.307
Apo-B/ApoA-I (ratio)	0.6 (0.5; 0.6)	0.6 (0.5; 0.7)	0.7 (0.5; 0.9)	0.081	1.000	0.110	0.207
Glucose (mg/dL)	93.0 (84.0; 97.3)	90.0 (83.0; 100.8)	103.0 (84.3; 100.5)	0.999	1.000	1.000	1.000
hs-CRP (mg/L)	0.5 (0.3; 0.6)	0.3 (0.2; 0.5)	0.4 (0.2; 0.9)	0.972	1.000	1.000	1.000
AST (Unit/L)	17.5 (13.0; 19.5)	17.0 (14.0; 20.8)	17.0 (13.3; 21.0)	0.861	1.000	1.000	1.000
ALT (Unit/L)	13.5 (9.0; 17.0)	12.0 (10.0; 16.0)	12.5 (11.0; 16.0)	0.828	1.000	1.000	1.000
γ-GTP (Unit/L)	11.0 (9.0; 14.0)	11.5 (8.3; 14.8)	15.0 (10.0; 33.5)	0.105	1.000	0.161	0.221

ApoA-I—apolipoprotein A-I; Apo-B—apolipoprotein B; AST—aspartate aminotransferase; ALT—alanine transaminase; HDL-C—high-density lipoprotein-cholesterol; hs-CRP—high-sensitivity C-reactive protein; LDL-C—low-density lipoprotein-cholesterol; γ-GTP—gamma-glutamyl transferase; SEM—standard error of the mean; TC—total cholesterol; TG—triglyceride. ^1^ Range of alcohol consumption amount (g) per month 21~105 g in group 2; ^2^ range of alcohol consumption amount (g) per month 168~1680 g in group 3. * Data are presented as median (25th; 75th percentile).

**Table 3 ijms-23-08623-t003:** Characteristics of lipoproteins from middle-aged women group depending on the alcohol intake *.

	Group 1 (0 g) *n* = 18	Group 2 (21~105 g) ^1^ *n* = 20	Group 3 (168~1680 g) ^2^ *n* = 16	*p*	1 vs. 2	1 vs. 3	2 vs. 3
LDL-size (nm^2^)	627 (487; 779)	494 (445; 571)	460 (390; 508)	<0.001	0.008	<0.001	1.000
LDL-MDA (nM)	1.2 (0.9; 1.5)	1.8 (1.3; 2.0)	1.5 (1.3; 1.9)	0.011	0.013	0.086	1.000
LDL-glycation (FI)	4236 (3825; 5007)	4489 (4116; 5434)	4568 (3632; 5547)	0.710	1.000	1.000	1.000
LDL-TC (mg/mg of protein)	19.7 (15.3; 21.8)	18.0 (15.1; 22.2)	17.4 (15.1; 22.9)	0.926	1.000	1.000	1.000
LDL = TG (mg/mg of protein)	2.5 (1.9; 3.4)	3.4 (2.9; 4.8)	3.8 (3.1; 4.3)	0.004	0.027	0.005	1.000
HDL2 -size (nm^2^)	214 (160; 241)	125 (98; 138)	183 (144; 200)	<0.001	<0.001	0.309	0.001
HDL2-glycation (FI)	1568 (1283; 1757)	1512 (1415; 1709)	1608 (1421; 1832)	0.510	1.000	0.834	1.000
HDL2-TC (mg/mg of protein)	2.8 (1.8; 3.3)	3.3 (2.1; 3.7)	3.5 (2.7; 3.6)	0.052	0.076	0.020	0.511
HDL2-TG (mg/mg of protein)	0.6 (0.4; 0.8)	0.5 (0.3; 0.7)	0.6 (0.5; 0.9)	0.328	0.530	0.388	0.136
HDL2-PON (Absorbance at 415 nm)	0.247 (0.175; 0.412)	0.183 (0.113; 0.261)	0.237 (0.128; 0.326)	0.021	0.007	0.048	0.582
HDL3-size (nm^2^)	111 (81; 142)	92 (79; 102)	81 (63; 100)	<0.001	0.031	<0.001	0.822
HDL3-TC (mg/mg of protein)	2.8 (2.4; 4.0)	3.1 (2.9; 4.0)	3.1 (2.9; 3.5)	0.463	0.741	1.000	1.000
HDL3-TG (mg/mg of protein)	0.3 (0.3; 0.4)	0.5 (0.4; 0.6)	0.6 (0.5; 0.9)	0.002	0.032	<0.001	0.118
HDL3-PON (Absorbance at 415 nm)	0.797 (0.661; 1.226)	0.941 (0.598; 1.070)	0.465 (0.274; 0.811)	0.010	1.000	0.055	0.012

FI—fluorescence intensity; HDL-C—high-density lipoprotein-cholesterol; LDL-C—low-density lipoprotein-cholesterol; MDA—malondialdehyde; PON—paraoxonase; SEM—standard error of the mean; TC—total cholesterol; TG—triglyceride. ^1^ Range of alcohol consumption amount (g) per month 21~105 g in group 2; ^2^ range of alcohol consumption amount (g) per month 168~1680 g in group 3. * Data are presented as median (25th; 75th percentile).

**Table 4 ijms-23-08623-t004:** Spearman rank correlation (r) and *p* values from linear regression between alcohol intake and the lipid parameters.

		r	*p*
Anthropometric profiles	BMI (kg/m^2^)	−0.135	0.329
	Fat mass_subcutaneous (kg)	−0.064	0.648
	Fat mass_visceral (kg)	−0.097	0.485
	SBP (mmHg)	−0.007	0.962
	DBP (mmHg)	0.152	0.272
	HS (kg)	−0.156	0.261
Blood lipid andinflammatory parameters	HDL-C (mg/dL)	−0.363	0.007
	ApoA-I (mg/dL)	−0.365	0.007
	ApoB (mg/dL)	0.107	0.440
	ApoB/ApoA-I (ratio)	0.281	0.040
	LDL-C (mg/dL)	0.087	0.533
	LDL-MDA (nM)	0.315	0.020
	TC (mg/dL)	0.025	0.860
	TG (mg/dL)	0.157	0.257
	TG/HDL-C (ratio)	0.222	0.107
	AST (Unit/L)	0.045	0.745
	ALT (Unit/L)	0.018	0.896
	γ-GTP (Unit/L)	0.237	0.084
	hs-CRP (mg/L)	−0.032	0.819
Lipoprotein profiles	LDL-TC (mg/mg of protein)	0.438	<0.001
	LDL-TG (mg/mg of protein)	0.016	0.907
	HDL_2_-TC (mg/mg of protein)	0.325	0.017
	HDL_2_-TG (mg/mg of protein)	0.153	0.270
	HDL_3_-TC (mg/mg of protein)	0.205	0.153
	HDL_3_-TG (mg/mg of protein)	0.500	<0.001
	HDL_3_-PON (AU)	−0.346	<0.001

ALT—alanine transaminase; AST—aspartate aminotransferase; ApoA-I—apolipoprotein A-I; ApoB—apolipoprotein B; AU—arbitrary unit; BMI—body mass index; DBP—diastolic blood pressure; γ-GTP—gamma-glutamyl transferase; HDL—high-density lipoprotein; HDL-C—high-density lipoprotein-cholesterol; HS—handgrip strength; hs-CRP—high-sensitivity C-reactive protein; LDL—low-density lipoprotein; LDL-C—low-density lipoprotein-cholesterol; MDA—malondialdehyde; PON—paraoxonase; SBP—systolic blood pressure; TC—total cholesterol; TG—triglyceride.

## Data Availability

The data used to support the findings of this study are available from the corresponding author upon reasonable request.

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
