# Peer review of "Long-Term Alcohol Consumption Caused a Significant Decrease in Serum High-Density Lipoprotein (HDL)-Cholesterol and Apolipoprotein A-I with the Atherogenic Changes of HDL in Middle-Aged Korean Women"

_ijms, 2022, doi:10.3390/ijms23158623_

Round 1
Reviewer 1 Report
The article is devoted to the study of the actual problem of the influence of long-term alcohol intake on the lipid-lipoprotein blood profile in women.
My comments:
1. You need to change the title of the article. For example: Long-term alcohol consumption caused a significant decrease in serum HDL-C and apoA-I with the atherogenic changes of HDL in middle-aged Korean women
2. The number of examined women is not large. I am sure that the distribution of indicators is non-parametric. Therefore, the data in tables 1-3 should be presented as Me [25%; 75%].
3. It is necessary to add reference links of the last 5 years (2017-2022) to the article.
4. There are spelling errors in the text, for example, in line 455.
Author Response
Please find attached doc file for response to Reviewer 1

Reviewer 2 Report
The revised manuscript, entitled “Comparison of high-density lipoprotein (HDL) qualities and functionalities between drinkers and non-drinkers from middle-aged Korean women: long-term alcohol consumption caused a significant decrease in serum HDL-C and apoA-I with the atherogenic changes of HDL ” by Kyung-Hyun Cho et al. is interesting but suffers from some flaws:
Comments:
1. The strong part of this research is a methodology used to HDL-C and LDL-C qualities and functionalities measurement.
2. The main problem of this study is a small sample size. It is well known that HDL-cholesterol levels are related to age and menopausal status and other confounding factors. The women in group 3 are younger and less menopausal (you did not observe statistical significance p=0,208, but this lack of significance is probably a result of small sample size). You should use logistic regression models to rule out confounding factors (like age, menopausal status etc.). Your sample size is too small to perform an adjusted logistic analysis to reach a convincing conclusions of this study. The conclusion in this study with such a small cohort is not convincing enough. Such small sample size would be more appropriate for a prospective study than for case-control study.
3. I think that it is incorrect to state that there is no data on long-term alcohol consumption in relation to metabolic syndrome component like HDL-cholesterol (line 79). You cited the reference from 2002 for this statement, Are you sure that there has been no research on this topic in 20 years. You should also include population-based studies performed in larger cohorts with adjustment for age, gender and other confounding factors like smoking, physical activity (e.g. DOI: 10.1017/S1368980020004449).
4. Description of participants and alcohol consumption is poorly described in the method section and inconsistent with tables [ e.g. 50±6 (line 432) vs 21~105 g in tables]
- You calculate the alcohol consumption per month (line 427). How long women had to actively drank alcohol to be long-term alcohol drinker?. In the abstract you mention “middle-aged women who consumed alcohol for at least 10 years of adulthood”. Please add this information to methods.
- WHO definition of alcohol consumption (line 413), please add reference. How you define moderate drinkers (line 15)?. In the methods you classified the women to mild drinkers and binge drinking.
- The alcohol intake was estimated from a self-administered questionnaire. Please precise the questions for alcohol intake in this questionnaire. How did you calculate the grams of alcohol per week or month (how many drinks per day, week or months?)?
5. The medians and interquartile ranges would be better for nonparametric statistics.
6. The title of this study is focused on HDL-cholesterol, but the aim and the final conclusion relate to HDL and LDL qualities?
7. Middle-aged women are usually defined as over the age of 40. Please add reference for the age range: 35–62 years old (line 421).
8. For nonparametric data you should use the Spearman rank correlation analysis.
9. Informed Consent Statement: Not applicable. Is it correct statement for clinical study?
10. “Moreover, there has been no study to explain the long-term effects of alcohol consumption for at least twenty years in middle-aged women regarding the change in HDL-C, HDL functionality, and qualities of HDL and LDL” line 322. You analyzed women who consumed alcohol for at least 10 years of adulthood?
Author Response
Please find attached doc for response to reviewer 2

Round 2
Reviewer 1 Report
The authors have done work to improve the quality of the article.
I have no more comments.
I believe that the article in its present form is worthy of publication in a journal.
Author Response
Thank you very much.
Your comments were really helpful to improve our paper
Reviewer 2 Report
The authors have done work to improve the quality of the article but I have a few more minor comments:
1. Your data have non parametric distributions. For that reason use medians instead of means. Please show your results as medians and interquartile ranges [(Me (25%; 75%)]. Please change the description of tables 1-3 [remove Mean±SEM; use Me (25%; 75%)]. Please change the description of statistical methods (page 12).
2. The information of small sample size should be included in the limitation section (page 8). Your aim is the comparison the quality of HDL and LDL among non-drinkers, mild drinkers, and binge drinkers, however the difference in menopausal status between groups interferes with the results.
3. The footnotes to indicate range of alcohol intake amount for group 2 and 3 (Range of alcohol consumption amount (g) per month 21~105 g in group 2 and Range of alcohol consumption amount (g) per month 168 ~1 680 g in group 3) should be also included under the table 2 and 3. This information should be also included in the method section (page 10 Participants).
Author Response
Thank you very much.
Your comments were helpful to improve our paper.
Please find attached doc for point-to-point our response and revision details
